# The Two Faces of Immune-Related lncRNAs in Head and Neck Squamous Cell Carcinoma

**DOI:** 10.3390/cells12050727

**Published:** 2023-02-24

**Authors:** Lesly J. Bueno-Urquiza, Marcela G. Martínez-Barajas, Carlos E. Villegas-Mercado, Jonathan R. García-Bernal, Ana L. Pereira-Suárez, Maribel Aguilar-Medina, Mercedes Bermúdez

**Affiliations:** 1Department of Physiology, University Center for Health Sciences, University of Guadalajara, Guadalajara 44340, Mexico; 2Faculty of Dentistry, Autonomous University of Chihuahua, Chihuahua 31000, Mexico; 3Department of Microbiology and Pathology, University Center for Health Sciences, University of Guadalajara, Guadalajara 44340, Mexico; 4Faculty of Biological and Chemical Sciences, Autonomous University of Sinaloa, Culiacán, Sinaloa 80030, Mexico

**Keywords:** HNSCC, tumor microenvironment, LncRNAs, cancer-associated fibroblasts

## Abstract

Head and neck squamous cell carcinoma (HNSCC) is a group of cancers originating from the mucosal epithelium in the oral cavity, larynx, oropharynx, nasopharynx, and hypopharynx. Molecular factors can be key in the diagnosis, prognosis, and treatment of HNSCC patients. Long non-coding RNAs (lncRNAs) are molecular regulators composed of 200 to 100,000 nucleotides that act on the modulation of genes that activate signaling pathways associated with oncogenic processes such as proliferation, migration, invasion, and metastasis in tumor cells. However, up until now, few studies have discussed the participation of lncRNAs in modeling the tumor microenvironment (TME) to generate a protumor or antitumor environment. Nevertheless, some immune-related lncRNAs have clinical relevance, since AL139158.2, AL031985.3, AC104794.2, AC099343.3, AL357519.1, SBDSP1, AS1AC108010.1, and TM4SF19-AS1 have been associated with overall survival (OS). MANCR is also related to poor OS and disease-specific survival. MiR31HG, TM4SF19-AS1, and LINC01123 are associated with poor prognosis. Meanwhile, LINC02195 and TRG-AS1 overexpression is associated with favorable prognosis. Moreover, ANRIL lncRNA induces resistance to cisplatin by inhibiting apoptosis. A superior understanding of the molecular mechanisms of lncRNAs that modify the characteristics of TME could contribute to increasing the efficacy of immunotherapy.

## 1. Introduction

Cancer, a group of multifactorial diseases, is considered one of the main public health problems, being the second cause of death worldwide [1]. According to GLOBOCAN, HNSCC incidence and mortality are about 800,000 and 400,000 cases, respectively, positioning it as the sixth most common cause of cancer death around the world [2].

HNSCC develops from squamous cells in the mucosal epithelium lining the oral cavity, larynx, oropharynx, nasopharynx, and hypopharynx [3,4]. This type of cancer is more common in men, with a 3:1 ratio compared with women [5], and occurs mainly after the age of 55 [6,7]. The main factors related to the development of this type of cancer are the consumption of alcohol and tobacco [8], whose effect is proportional to the intensity of exposure [9].

Additionally, it has been described that infection with high-risk human papillomavirus (HPV), mainly genotypes 16, 18, 31, 33, and 35, acts synergistically in carcinogenesis. In this regard, HNSCC can be classified as HPV-negative and HPV-positive [10,11,12]. HPV infection is responsible for up to 60% of HNSCC cases, as it participates in the development of oropharyngeal tumors, being the 90% of HPV-positive tumors related to HPV 16 infection. Interestingly, HPV infection, in addition to being an etiological factor, is related to the prognosis of patients. It has been observed that HPV-positive cases show a favorable prognosis, unlike those that are not [10,12].

### Tumor Microenvironment

The TME is a very complex construct composed of extracellular matrices (ECM) and cellular components such as tumor cells, immune cells, and cancer-associated fibroblasts (CAFs), among others [13,14]. For this, tumors can be classified according to the cellular infiltrate as inflamed tumors and immune-excluded and immune-desert tumors. Inflamed tumors are characterized by abundant intratumoral and stromal immunological infiltrate. Immune-excluded tumors have immunological infiltrate restricted to the stroma. Immune-deserts lack infiltrate both in the tumor and in the stroma [15].

Even though in inflamed tumors there is an infiltrate of immune cells, in an immunosuppressive environment the tumor can evade the host response and progress [15,16]; this also depends on the infiltrate and its relationship with a positive or negative prognosis. The most common model to explain the tumor behavior is “cancer immunoediting”, which refers to a dual action that the immune system can take, one of which is the protection towards the host by eliminating tumor cells, the other is the programming of, those cells of the immune system that are associated with the tumor and help tumor progression [17]. This process can be divided into three phases that are called the “three E” (elimination, equilibrium, and escape). First, elimination refers to immunosurveillance mediated by the immune cells; second, equilibrium is where the immune system promotes the generation of tumor cells that survive the attack; finally, once immunological anergy and tolerance are achieved, escape leads to cancer cells that can form tumors [17,18].

The immune infiltrate in TME includes cells from the adaptive immunity such as cytotoxic T lymphocytes (CTL) that recognize and kill tumor cells through the release of granzymes and perforins, CD4+ T cells that are essential for the proliferation and differentiation of CD8+ T cells that infiltrate the tumor, innate immune cells such as natural killer (NK) cells that have cytotoxic and cytokine-producing activity, tumor-associated macrophages (TAMs) classified into two subpopulations (M1 with antitumor activity and M2 with protumor activity and an immunosuppressive profile), mast cells that release preformed inflammatory mediators in their granules, and finally stromal cells such as CAFs that are fibroblasts functionally different from the normal population and participate in the remodeling of the extracellular matrix and the production of protumoral cytokines [15,16,19,20,21,22].

The antitumor immune response is characterized by an infiltrate of CTL, B lymphocytes, CD4+ Th1 lymphocytes, regulatory T cells (Treg), M1 macrophages, and NK cells, while CD4+ Th2 lymphocytes, M2 macrophages, neutrophils, and CAFs, among others, participate in the protumoral immune response [16,19,20]. These cell populations have intercellular communication through cytokines, chemokines, and non-coding RNAs (ncRNAs) [22,23,24,25], which will modulate the characteristics of TME [26]. ncRNAs represent a large percentage of the genome with relevant functions in biological processes since they control the expression of genes. ncRNAs can be classified according to their length in microRNAs (miRNA), which have a length of approximately 22 nucleotides, and the lncRNA, which are longer than 200 nucleotides [25,27].

## 2. Long Non-Coding RNAs

LncRNAs are non-coding chains of 200 to 100,000 nucleotides transcribed by RNA polymerase II [28]. Generally, they have a poly-A tail and can be subjected to splicing processes [27,29]. Their mechanisms of action are diverse both in the cytoplasm and in the nucleus. In the cytosol, they are related to the regulation of mRNA decay as well as its stability, functioning as sponges for miRNAs. Meanwhile, in the nucleus, they are associated with promoter sites, participating in transcriptional repression, epigenetic regulation, and nuclear architecture [30,31].

LncRNAs play an important role in both innate and adaptive immune responses; it has been shown that they affect essential processes such as differentiation or the immune function [11,32,33]. It has been observed that some of the biological processes they regulate are cell activation, proliferation, metabolism, and death [28,34].

## 3. LncRNAs in HNSCC Tumor Microenvironment

### 3.1. Tumor Cells

In recent years, the participation of lncRNAs in the tumorigenesis of cancer cells involving the tumor microenvironment has gained relevance given that some of the lncRNAs are associated with poor prognosis (Table 1). In this regard, the lncRNA MIR31HG is associated with poor prognosis since its expression is significantly correlated with advanced stages in laryngeal squamous cell carcinoma (LSCC) samples and in vitro and in vivo analysis found that it promotes cancer cell growth [35]. In addition, USP2-AS1 promotes progression through proliferation, tumor growth, invasion, and the transition from G0/G1 to the S phase of the cell cycle in both in vitro and in vivo models [36]. On the other hand, the lncRNA TM4SF19-AS1 acts as a sponge for miR-153-3p since it binds to LAMC1 (laminin gamma 1 subunit), which has been reported to be upregulated in patients with HNSCC [37]; thus, TM4SF19-AS1 enhances proliferation, migration, invasion, and epithelial–mesenchymal transition (EMT) through the expression of mesenchymal markers (vimentin, N-cadherin) [38].

Furthermore, LINC00460 is associated with the regulation of proliferation, migration, invasion, and mesenchymal marker expression in vitro [40]. In the case of HCG18, lncRNA is overexpressed in cell lines and patients with HNSCC regulating migration, invasion, and modulating progression through the expression of cyclin D, which is a key protein in the WNT signaling pathway and is directly associated with a poor prognosis of patients [41].

The lncRNAs not only act in the progression of cancer but also in tumor suppression, being associated with a good prognosis. For instance, HNSCAT1 is downregulated in samples of advanced HNSCC, meanwhile its overexpression is associated with the formation of minor tumors in vivo [43].

### 3.2. CAFs

Due to the heterogeneity of CAFs, several pathways participate in their activation. Recently, the role of some lncRNAs that participate in the modulation of their activation has been described as finding the stimulus with the factor PDGF-BB (platelet-derived growth factor-BB), associated with differentiation towards CAFs [67], also increases the expression of the lncRNA LURAP1L-AS1 (leucine-rich adaptor protein 1-like antisense RNA 1) as well as the classical markers of CAFs (α-SMA (α-smooth muscle actin), FSP-1 (fibroblast-specific protein 1), and FAP (fibroblast activation protein)).When LURAP1L-AS1 silencing is performed, the expression of the markers decreases; it also participates in the regulation of NF-κB through the LURAP1L-AS1/LURAP1L/IKKa/IκBa/NF-κB axis [44]. Another lncRNA overexpressed is FLJ22447 or lncRNA-CAF that, in conjunction with IL-33, participates in NF activation toward CAFs. LncRNA-CAF silencing has an impact on the decreased expression of classical CAF markers and lncRNA-CAF functions as a lncRNA scaffold to maintain IL-33 protein stability and inhibit its degradation [45].

Recently, the lncRNA LOC100506114 was found to be overexpressed in the tumor stroma, indicating that expression is driven by mesenchymal cells. Subsequently, increased expression of LOC100506114 was found in CAFs isolated from patients in comparison with NF [46]. Furthermore, functional analysis on tumor cells co-cultured with CAF-conditioned medium determined the increase in migration, proliferation, and expression of mesenchymal markers. Briefly, the studies showed that growth differentiation factor 10 (GDF10) promotes the functional transformation of an NF to a CAF via LOC100506114 that binds to the transcription factor RUNX2, which, in turn, participates in tumor growth, invasion, and metastasis [46].

Some lncRNAs participate in the regulation of glucose metabolism of oral CAFs, as reported by Yang et al., where lncRNA H19 was identified to modulate glucose metabolism [47]. When its expression is suppressed, it decreases glucose uptake and lactate secretion. It also regulates fundamental processes such as proliferation and migration. It has been reported that H19 exerts sponge or precursor functions of various miRNAs. In this case, it was reported to be a precursor of Hsa-miR-675 that interacts with the PFKFB3 gene in the glycolysis pathway in oral CAFs [48].

Important processes such as angiogenesis and metastasis are regulated by lncRNAs in CAFs. However, they can result in a better or worse prognosis for patients, depending on the regulation at the gene level. For instance, patients who overexpress FOXF1 adjacent noncoding developmental regulatory RNA (FENDRR) have a better prognosis, because, when it is overexpressed, there is less migration in vitro and it also can regulate proangiogenic activity through the PI3K/AKT pathway [66]. Conversely, a lncRNA associated with a poor prognosis is the new one called TIRY, which indirectly regulates cancer cells due to the effect of CAF-conditioned medium on tumor cells, where it was reported that TIRY is upregulated and facilitates increased invasion, migration, and metastasis in addition to acting as a miRNA sponge of miR-14 and inducing activation of the WNT/b-catenin pathway resulting in increased EMT [49].

There are molecules secreted by CAFs that regulate the expression of lncRNAs in tumor cells as reported by Zhang et al., reporting that the Midkine molecule (MK) secreted by the tumor stroma regulates the expression of the lncRNA ANRIL and participates directly in the resistance to cisplatin, showing that CAF-conditioned medium in stimulated cancer cells induces cisplatin resistance, thus suggesting that the MK secreted by CAFs in a paracrine manner towards tumor cells regulates the resistance to cisplatin by inhibiting apoptosis [68].

The use of lncRNAs as therapeutic targets has gained relevance in recent years since they could act in response to chemotherapy. For instance, it has been reported that, when lncRNA IL7R is silenced and a TLR3 inhibitor is used, tumor cells are more sensitive to treatment and apoptosis increases in epithelial cells cocultured with CAFs, in addition to increasing the immune infiltrate with immune cells associated with a better prognosis such as dendritic cells and CD8+ lymphocytes [51].

### 3.3. Immune-Related lncRNAs

In TME, tumor cells interact with other cell populations such as CAFs, endothelial cells, and cells of the immune system [33] through complex communication networks, enhancing tumor modulation of the microenvironment. Thus, TME plays an essential role in the initiation, tumor growth, invasion, and metastasis (Figure 1). In addition, the HNSCC TME is highly infiltrated by immune cells, which, depending on tumor biology, may mediate immune surveillance or evasion through various mechanisms [3]. Recently, increasing evidence has revealed that lncRNAs regulate the immune response in TME by controlling the type of cellular infiltration, differentiation, and functions of immune cells [32,69], which can suppress or favor the progression of cancer. Hence, the study of the involvement of immune-related lncRNAs on the evolution of HNSCC has gained importance.

Recent studies have identified immune-related lncRNAs in HNSCC impacting the prognosis of patients. Using bioinformatic tools, Chen et al. selected seven immune-related lncRNAs associated with OS: AL139158.2, AL031985.3, AC104794.2, AC099343.3, AL357519.1, SBDSP1, and AC108010.1. With these lncRNAs, they built a prognostic signature and classified HNSCC patients as low- or high-risk. Furthermore, they identified that low-risk cases have a more significant infiltration of immune cells and enrichment of pathways associated with the immune response. In contrast, high-risk cases are related to the enrichment of metabolic pathways [70]. This result is consistent with previous reports that identified nine immune-related lncRNAs in nasopharyngeal carcinoma, where low-risk patients have active pathways associated with the immune response and a greater intratumoral infiltrate of CD8+ T cells and B cells. In contrast, in high-risk patients, there is an association with pathways involved in metabolism [71].

In the case of OSCC, previous research divided samples according to the expression of eight ferroptosis-related lncRNAs with implications in the prognosis. In the low-risk group, a significant decrease in AL512274.1, MIAT, and AC079921.2 was found, related to a more intense immune response compared with the high-risk group, where the expression of FIRRE, AC099850.3, and AC090246.1 increased [72]. Multiple reports relate the cases of HNSCC that present a better prognosis with an active immune response, which can be associated with an abundant infiltrate of immune cells [33]. However, there are tumor characteristics that can modify the expression of specific immune-related lncRNAs and, with this, induce an immunosuppressive TME.

Mutations in the tumor suppressor genes *TP53* and *CDKN2A* are frequent in HNSCC, and tumor cells that present these modifications can alter their pattern of expression of lncRNAs. At the same time, conditions such as hypoxia induce the expression of specific lncRNAs in immune cells. This crosstalk between tumor cells and immune cells induces the formation of an immunosuppressive TME [73]. On the other hand, it has been observed that there are lncRNAs expressed in tumor cells that promote immune activation. PRINS, an overexpressed lncRNA in some cases of HPV-positive HNSCC, is related to the activation of genes involved in the immune response. Among HPV-positive tumors, those with higher PRINS levels are associated with a better prognosis [52]. Due to this, research is focused on studying how lncRNAs regulate the differentiation and function of specific populations of immune cells in TME, which ultimately impacts tumor progression.

Bioinformatic analysis of HNSCC samples has allowed the identification of various lncRNAs associated with genomic instability [74] or with the immune response [75,76] and related to the prognosis of patients for grouping the cases in low and high risk. Furthermore, since the inflammatory infiltrates in TME can exert a dual anti- or protumor function, the types of immune cell populations that infiltrate both groups of tumors have been studied. Different reports agree that in the low-risk group, there is a greater infiltrate of activated CD8+ T cells, activated CD4+ T cells, T follicular helper (T fh) cells [20,77], Treg cells [20,76], NK cells, B cells [74], and resting mast cells [76], as well as decreased numbers of M0 macrophages, activated mast cells [20], and CAFs [74]. In contrast, the high-risk group is characterized by increased eosinophil infiltration, naive CD4+ T cells, resting NK cells, M0 macrophages [76], M2 macrophages [78], activated mast cells [76], and CAFs [74], accompanied by the decrease in the expression of human leukocyte antigen (HLA) molecules [76] necessary to sustain the activation of the immune response. These findings show that the cell populations in low-risk cases are associated with an anti-tumor immune response, whereas an immunosuppressive microenvironment predominates in high-risk tumors. For a better understanding of the factors involved in the modulation of TME characteristics, the role played by some lncRNAs in the differentiation or polarization of immune cells has been studied.

In LSCC, a considerable infiltrate of M2 macrophages plays a protumor role. HOTAIR is a lncRNA expressed in LSCC tumor cells and can be released into exosomes, which is related to M2 macrophage polarization via the downregulation of *PTEN* and the upregulation of PI3K and AKT expression. In in vitro analysis, the co-culture of M2 macrophages polarized with exosomes and LSCC cell lines increased the proliferation and migration of tumor cells. Interestingly, the in vivo injection of exosome-treated macrophages promoted an increase in tumor size, downregulation of the epithelial marker E-cadherin, and increased expression levels of the mesenchymal marker N-cadherin related to the EMT [54]. Regarding M1 macrophages, with bioinformatic analysis of databases, it has been identified that in OSCC, LINC00460 is positively correlated with this cell phenotype and CASC9 is negatively correlated; both have a strong correlation with the prognosis of patients [75].

Within the group of innate immune cells present in TME are neutrophils, the study of which has gained importance in recent years due to the impact that the functions of these cells have on tumor progression [79]. In an evaluation of the lncRNAs associated with NETosis (formation of neutrophil extracellular traps (NETs)), in HNSCC, it was found that low-risk patients present enrichment of pathways associated with the immune response. At the same time, high-risk cases correspond to cold tumors associated with NETosis activation. Among the lncRNAs identified, LINC00426 is a protective factor. When nasopharyngeal carcinoma cell lines are transfected with this lncRNA, its overexpression significantly increased the expression levels of p-STING, p-TBK1, and p-IRF3. In addition, activation of the STING signaling pathway promotes the secretion of cytokines necessary for the recruitment of T cells and B cells, such as CXCL10, CCL5, ISG15, and ISG56 [11].

MANCR is highly expressed in HNSCC tissue and cell lines, related to poor OS and disease-specific survival. In vitro, MANCR silencing inhibits the proliferation, migration, and invasion of HNSCC cell lines. However, in addition to acting as an oncogene, bioinformatic analyses have revealed that its expression is positively correlated with the infiltrate of neutrophils and γδ T cells but negatively with the presence of CD8+ T cells and B cells [55]. The type of inflammatory infiltrate is modulated by the cytokines secreted in the TME; this cytokine secretion can, in turn, be affected by the expression of lncRNAs. For example, BARX1-DT, KLHL7-DT, and LINC02154 are expressed in LSCC. These immune-related lncRNAs can promote an immunosuppressive TME by decreasing the expression of CCR3, CXCL9, and CXCL10, decreasing the recruitment of CD8+ T cells [56].

There are immune-related lncRNAs that, in addition to being involved with the secretion of cytokines, are also related to the expression of other molecules necessary to mount the antitumor immune response. For example, TRG-AS1 is expressed in warm tumors with a high infiltration of cytotoxic cells, related to a better prognosis. It has been shown in vitro that the silencing of TRG-AS1 in an OSCC cell line suppresses the expression of HLA-A, HLA-B, and HLA-C molecules necessary for antigen presentation, as well as CXCL9, CXCL10, and CXCL11 [32]. On the other hand, LINC02195 has high expression in the nucleus and cytoplasm of HNSCC cells, which is associated with a good prognosis as it has a positive correlation with the infiltrate of CD4+ T cells and CD8+ T cells, in addition to being involved in the expression of the MHC-I, antigen processing, and presentation [58].

The set of molecules that regulate the immune response includes costimulatory and coinhibitory molecules that are also targets for modification by lncRNAs. IFITM4P progressively increases its expression from premalignant lesions such as OL to OSCC, thus acting as an oncogene. In a murine model of carcinogenesis in the tongue, lipopolysaccharides (LPS) bind to its receptor TLR4, which induces an increase in the expression of IFITM4P, acceleration of the carcinogenesis process, and immune escape through overexpression of the PD-L1 immunoregulatory ligand. IFITM4P induces PD-L1 expression in two different ways. In the cytoplasm it acts as a scaffold for the recruitment of SASH1, which binds and phosphorylates TAK1; this increases NF-κB phosphorylation, which ultimately induces PD-L1 expression. In the nucleus, IFITM4P reduces the transcription of *PTEN* by increasing the binding of KDM5A to its promoter and, with this, it upregulates PD-L1. In contrast, the overexpression of IFITM4P increases the sensitivity to treatment with PD-1 mAb [59].

LncRNAs are associated with tumor immune evasion. LINC01123 is overexpressed in HNSCC tissue and cell lines, mainly in the cytoplasm of the cells, which, together with the overexpression of the immune checkpoint B7-H3, is associated with a poor prognosis by promoting tumor immune evasion. Furthermore, LINC01123 is competitively bound to miR-214-3p, and miR-214-3p, specifically targeting *B7–H3*; this inhibits CD8+ T cell activation and favors tumor progression. By silencing LINC01123 in HNSCC cell lines, the cytotoxic activity of CD8+ T cells increased, thereby decreasing tumorigenicity and increasing the secretion of factors associated with immune activation in vivo [60].

LncRNAs also participate in sculpting the TME and include the activation or inhibition of specific pathways. For example, LINC01355 is overexpressed in OSCC and is associated with antitumor evasion by inhibiting the activity of CD8+ T cells through activation of the Notch pathway. Conversely, by deleting LINC01355 in OSCC cells, apoptosis of CD8+ T cell is retrained, proliferation and cytolysis activity is enhanced, and tumor cell proliferation, migration, and invasion are decreased [61].

The expression of lncRNAs in TME immune system cells has been less studied. However, the reported evidence shows that they have an impact on tumor progression due to the bidirectional communication that exists between tumor cells and stromal cells. DCST1-AS1 is overexpressed in OSCC tumor cells and M2 macrophages. Silencing this lncRNA has been shown in vitro and in vivo to block NF-κB signaling, therefore repressing tumor cell emergence, migration, and invasion, as well as protumor M2 polarization of macrophages [62].

In the case of CRNDE, it is expressed in OSCC, mainly in advanced stages in tumor cells and tumor-infiltrating T lymphocytes (TILs). Its expression in cancer cells exerts a protumor function by sponging miR-545-5p, which leads to increased expression of the immune checkpoint TIM-3 and suppresses the cytotoxicity of CD8+ T cells by contributing to their depletion [63]. In a mouse model, injecting CD8+ T cells with CRNDE-knockdown decreases tumor size, increases the number of IFN-γ and TNF-α-producing CD8+ T cells, decreases TIM-3 expression, and increases miR expression -545-5p, activating the antitumor immune response of CD8+ T lymphocytes [63].

Finally, HOTTIP is a lncRNA expressed by HNSCC tumor cells and present in the exosomes of M1 macrophages. Although it has been associated with a protumor function, one study reported that exosomes from M1 macrophages, primarily through HOTTIP, inhibit HNSCC progression by activating the TLR5/NF-κB signaling pathway by competitively sponging miR-19a-3p and miR-19b-3p. In addition, they polarize circulating monocytes and TAMs toward an antitumor M1 phenotype, inducing positive feedback [65].

### 3.4. LncRNAs: Therapeutic Targets and Clinical Relevance in HNSCC

Despite significant advances in the treatment of HNSCC, the mortality rate remains around 50% [32]. It is essential to explore new therapeutic strategies to improve patients’ time and quality of life. Lately, immunotherapy has received rising attention in cancer treatment for the OS advantages it offers; however, the overall response rate to immunotherapy in patients with HNSCC is less than 20% [80]. The understanding of the molecular mechanisms that modify the characteristics of the TME can contribute to the detection of lncRNAs as novel biomarkers to provide new ideas for clinical diagnosis, immune-targeted therapy, and drug discovery [33]. For example, identifying that HOTTIP polarizes circulating monocytes towards an antitumor M1 phenotype and suppresses HNSCC progression through the upregulation of the TLR5/NF-κB signaling pathway may provide novel insight into HNSCC immunotherapy [65].

LncRNAs may function as potential therapeutic targets (Figure 2), as it has been reported that, when their lnc-IL7R function is suppressed, there is better sensitivity to chemotherapy in oral cancer cell lines [51]. The mechanism by which IFITM4P induces PD-L1 expression is known, so this lncRNA may serve as a new therapeutic target in the blockage of oral carcinogenesis [59]. However, for most of the lncRNAs that show alteration in expression in HNSCC, the exact mechanism by which TME conditions are modified remains to be unknown; for this reason, more studies are required to clarify this information to develop new therapeutic strategies [20].

Hereby, we present data that show that some immune-related lncRNAs have clinical relevance, since AL139158.2, AL031985.3, AC104794.2, AC099343.3, AL357519.1, SBDSP1 [70], and AC108010.1 TM4SF19-AS1 [39] have been associated with overall survival (OS). MANCR [55] is also related to poor OS and disease-specific survival. MiR31HG [35], TM4SF19-AS1 [39], and LINC01123 [60] are associated with poor prognosis. Meanwhile, LINC02195 [58] and TRG-AS1 [32] overexpression is associated with favorable prognosis. Moreover, ANRIL [68] lncRNA induces resistance to cisplatin by inhibiting apoptosis. A superior understanding of the molecular mechanisms of lncRNAs that modify the characteristics of TME could contribute to increasing the efficacy of immunotherapy.

## 4. Conclusions

LncRNAs involved in TME are clinically relevant, being indicators of survival, acting in important processes such as chemoresistance and being indicators of prognosis. The study of lncRNAs in cancer can contribute to a better understanding of the molecular mechanisms that modify the characteristics of TME, allowing the detection of possible therapeutic targets and biomarkers that contribute to the best selection of patients who are candidates for immunotherapy, resulting in the increase in efficacy of this type of treatment in HNSCC.

## Figures and Tables

**Figure 1 cells-12-00727-f001:**
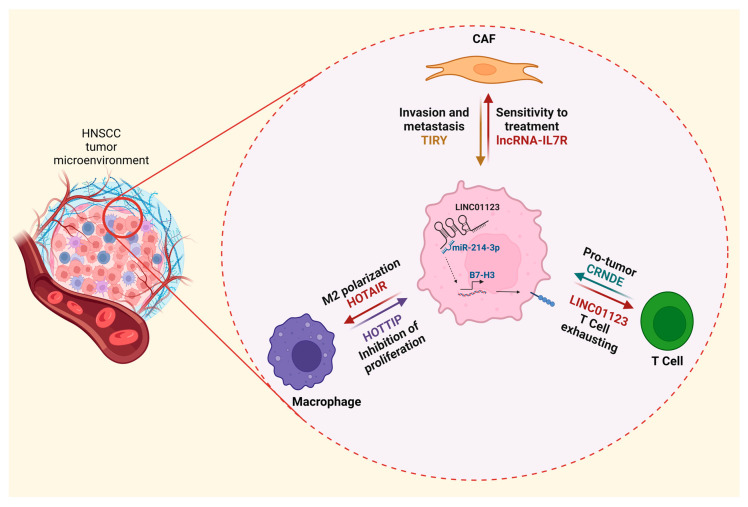
Interactions in the TME are mediated by crosstalk between cells of the immune system, stromal cells, and tumor cells, primarily regulating the behavior of the immune system to induce tumor escape. The image shows examples of crosstalk between some cellular components of the TME through communication with lncRNAs and the effect that is observed.

**Figure 2 cells-12-00727-f002:**
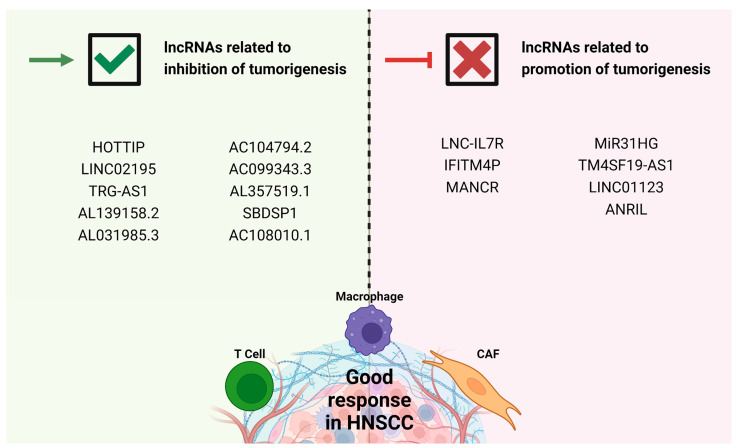
Potential lncRNAs that could be targeted. In the green background the lncRNAs associated to inhibition of tumorigenesis are listed; the therapeutic strategies could be directed to induce them within the TME. In the pink background there is a list of lncRNAs related to promotion of tumorigenesis in HNSCC; the therapeutic strategy could be directed to inhibit their expression within the TME.

**Table 1 cells-12-00727-t001:** Overview of lncRNAs in the HNSCC tumor microenvironment.

lncRNA	Status of Expression	Model	Participation in HNSCC
MiR31HG	Upregulated	LSCC cancer tissue	Plays an oncogenic role and its overexpression can serve as a poor prognosis marker [35].
USP2-AS1		In vitro model (HNSCC cell lines)	Inhibits cellular senescence, acts as an oncogenic molecule, and promotes progression through proliferation, tumor growth, and invasion [36].
TM4SF19-AS1		In vitro model (HNSCC cell lines), RNA sequencing dataset	Acts like sponge del miR-153-3p [38], associated with OS and prognosis [39].
cLINC00460		HNSCC tissues and in vitro model (HNSCC cell lines)	Regulates cancer progression and mesenchymal marker expression in CAFs [40].
HCG18		Tissue samples of HNSCC, HNSCC cell lines, and xenograft model/in vitro model (laryngeal and hypopharyngeal squamous cell carcinoma cell lines)	Promotes cell proliferation and metastasis and modulates progression through the WNT signaling pathway [41,42].
HNSCAT1		In vitro model of primary keratinocytes	Overexpression of HNSCAT1 significantly inhibited tumor progression through HNSCAT1 interaction with miR-1254 [43].
LURAP1L-AS1		In vitro model of oral fibroblasts	Activation of the canonical NF-κB pathway, inducing the transformation of NFs (normal fibroblasts) into CAFs [44].
FLJ22447/lncRNA-CAF		In vitro model of oral squamous cell carcinoma (OSCC) (primary culture of CAF) and OSCC cell line	Regulate IL-33 levels and prevented p62-dependent autophagy–lysosome degradation of IL-33 [45].
LOC100506114		In vitro model of OSCC (primary culture of CAFs)	Regulates fibroblast activation and promotes OSCC cell proliferation and migration through activation of TGFbR1/X2 and migration through activation of the TGFbR1/Smad3/ERK pathway of OSCC cells [46].
H19		In vitro model of OSCC (primary culture of CAFs)	Regulates the expression of enzymes, regulatory molecules, and oncogenes and/or oncogenes that indirectly modulate pathways involved in glucometabolic processes [47,48].
TIRY		In vitro model of OSCC (primary culture of CAFs)	It acts as a miRNA sponge and downregulates miR-14 expression, promoting invasion and metastasis through WNT-β-catenin activation in oral cancer cells [49].
ANRIL		In vitro model (OSCC cell lines)	Encodes 3 tumor-suppressor proteins, p15INK4b, p14ARF, and p16INK4a; its transcription is a key requirement for replicative or oncogene-induced senescence and constitutes an important barrier for tumor growth [39,50].
LncRNA-IL17R		In vivo model of OSCC	Regulate response to chemotherapy, and cancer progression [51].
PRINS		HNSCC RNA sequencing datasets	High expression in HPV-positive patients is associated with better OS. Is involved in the immune mechanisms, in mounting an antiviral response by affecting some pattern recognition receptors (PRRs) [52].
HOTAIR		Tissue samples; in vitro and in vivo models of LSCC	Highly expressed in the advanced clinical stages of LSCC [53]. Exosomal HOTAIR induces macrophages to M2 polarization by PI3K/p-AKT/AKT signaling pathway and these M2 macrophages facilitate the migration, proliferation, and EMT of LSCC in vitro and in vivo [54].
MANCR		Tissue samples and in vitro model of HNSCC	Is a high-risk factor in patients with HNSCC. Is associated with peripheral nerves and the extracellular matrix for highly expressed genes and hence may play a crucial role in the occurrence of HNSCC [55].
BARX1-DT KLHL7-DT LINC02154		RNA sequencing datasets and in vitro model of LSCC	Patients with LSCC and high expression of BARX1-DT [56], KLHL7-DT, and LINC02154 [56,57] have worse OS. These lncRNAs may boost the development of an immunosuppressive TME by downregulating the expression of key immunomodulators such as CCR3, CXCL10, and CXCL9 and subsequently decreasing the recruitment of effector CD8+ T cells [56].
TRG-AS1		RNA sequencing datasets and in vitro model of HNSCC	The high expression indicates a favorable prognosis in HNSCC. Is an essential lncRNA involving TME formation. Knockdown of TRG-AS1 inhibited the expression of HLA-A, HLA-B, HLA-C, CXCL9, CXCL10, and CXCL11 in vitro [32].
LINC02195		RNA sequencing datasets, tissue samples, and in vitro model of HNSCC	There is a correlation between high LINC02195 expression and favorable prognosis in HNSCC. Is associated with genes encoding MHC-I molecules, antigen processing, and presentation and is related to an increased number of CD8+ and CD4+ T cells [58].
IFITM4P		Oral leukoplakia (OL) and OSCC tissue samples, in vitro and in vivo models of OL and HNSCC	Acts as a scaffold to facilitate the recruitment of SASH1 to bind and phosphorylate TAK1 and further increase the phosphorylation of NF-κB to induce PD-L1 transcription, hence promoting immune evasion [59].
LINC01123		Tissue samples, in vitro and in vivo models of HNSCC	High expression is associated with poor prognosis in patients with HNSCC. Acts as a miR-214-3p sponge to inhibit the activation of CD8+ T cells and promote tumor immune escape by upregulating B7–H3 [60].
LINC01355		In vitro and in vivo models of OSCC	Could induce the development of OSCC via modulating the Notch signal pathway that represses CD8+ T cell activity [61].
DCST1-AS1		In vitro and in vivo models of OSCC	Contributes to cancer progression by enhancing the NF-κB signaling pathway to promote OSCC development and M2 macrophage polarization [62].
CRNDE		Tissue samples, in vitro and in vivo models of OSCC	The expression is higher in stage IV of OSCC than early stages. Can exhibit a crucial role in activating CD8+ T cell exhaustion by sponge miR-545-5p to induce TIM-3 expression [63].
HOTTIP		RNA sequencing datasets, in vitro/in vivo models of HNSCC	Is highly expressed in stages III-IV of HNSCC [64]. Overexpression of HOTTIP inhibits HNSCC progression and induces the polarization of M1 macrophages because it activates the TLR5/ NF-κB signaling pathway by competitively sponging miR-19a-3p and miR-19b-3p [65].
FENDRR	Downregulated	In vitro model of OSCC (primary culture of CAF)	Downregulation of FENDRR can activate the PI3K/AKT pathway in NFs and increases matrix metalloproteinase 9 (MMP9) expression [66].
LINC00426	Downregulated in nasopharyngeal carcinoma cell lines CNE1, HNE1, and TW03	HNSCC RNA sequencing datasets and in vitro model with nasopharyngeal carcinoma cell lines	Contributes to the innate immune cGAS-STING signaling pathway, related to the secretion of cytokines to recruit B cells and T cells, and promoting immune cell infiltration [11].

## Data Availability

Not applicable.

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
