# Peer review of "The Two Faces of Immune-Related lncRNAs in Head and Neck Squamous Cell Carcinoma"

_cells, 2023, doi:10.3390/cells12050727_

Round 1
Reviewer 1 Report
The authors present a comprehensive review of the roles of lncRNAs in head and neck squamous cell carcinoma (HNSCC) tumor progression and inhibition, and in the immune responses to these malignancies. The paper adequately describes the complex interplay between the tumor and immune cells in the tumor microenvironment. This work is important because as the authors point out, the survival of HNSCC has remained at 50% overall for the past 5 decades, and immunotherapy is effective in only 20% of these cancers. However, HNSCC is not a single disease, and survival is related to subsite. Table 1 summarizes the text but could be improved by adding a column indicating the head and neck subsites associated with each lncRNA described.
Author Response
We acknowledge editors and reviewers for their kind observations regarding our manuscript. All your recommendations have been followed and answered. Changes in the manuscript can be reviewed in the track changes of Word.
Moderate English changes required.
We appreciate your observations about language editing, the paper has been extensively reviewed and corrected as recommended by Desai et al,. Fluency and grammar have been reviewed and corrected. Missing prepositions and articles were also added improving the text in general. All changes can be seen along the text in Word track changes: “Desai G. How to leverage the language: A guide for medical writers in India. Perspect Clin Res. 2019 Jan-Mar;10(1):37-41. doi: 10.4103/picr.PICR_129_18. PMID: 30834207; PMCID: PMC6371706”. Besides, a thorough textual revision was done by a native English speaker to improve the document
HNSCC is not a single disease, and survival is related to subsite. Table 1 summarizes the text but could be improved by adding a column indicating the head and neck subsites associated with each lncRNA described.
We appreciate your comments, and clarification that HNSCC is a group of cancers has been done. Regarding the column of the table that is suggested to be added, we add a column regarding the model used in every study making clear the specific subsite and its clinical relevance
Reviewer 2 Report
The review entitled “The two faces of immune-related lncRNAs in Head and Neck Squamous Cell Carcinoma“ by Lesly J. Bueno-Urquiza and colleagues described the current state of research in lncRNA in HNSCC with focus on molecular mechanisms in tumor microenvironment with possible contributing effects regarding immunotherapy.
Detailed comments to this manuscript have been reported below.
1) Line 47: Please provide a citation.
2) Line 61-63: A long and not focused sentence. Please rephrase.
3) Line 88: lncRNA acronym needs to be explained in the main text before it is introduced
4) Line 176-186: Very long and convoluted sentences. Please rephrase.
5) Line 349-352: Again, a long and convoluted sentence. Please rephrase.
6) Regarding Table 1: Some references are not relevant to HNSCC because other cancer entities were researched in the References or the relevant lncRNAs are not mentioned. Please recheck the references and modify the relating lncRNA descriptions. Details below:
· Reference 39: Study with breast cancer (and not HNSCC)
· Reference 40: Study without any HNSCC cell lines
· Reference 41: No mentioning of ANRIL and regarding Reference 51: This review does not mention HNSCC
· Reference 53: Study with cervical cancer
· Reference 56: Study without HNSCC samples
· Reference 66: Study with beast epithelial cells and no link with LINC01355
Author Response
We acknowledge editors and reviewers for their kind observations regarding our manuscript. All your recommendations have been followed and answered. Changes in the manuscript can be reviewed in the track changes of Word.
Moderate English changes required.
We appreciate your observations about language editing, the paper has been extensively reviewed and corrected as recommended by Desai et al,. Fluency and grammar have been reviewed and corrected. Missing prepositions and articles were also added improving the text in general. All changes can be seen along the text in Word track changes: “Desai G. How to leverage the language: A guide for medical writers in India. Perspect Clin Res. 2019 Jan-Mar;10(1):37-41. doi: 10.4103/picr.PICR_129_18. PMID: 30834207; PMCID: PMC6371706”. Besides, a thorough textual revision was done by a native English speaker to improve the document
Line 47: Please provide a citation.
Well noted, thanks for your comment. Citation has been included within the text.
Line 61-63: A long and not focused sentence. Please rephrase.
Thanks for your kind observation. Paraphrasis has been done within the text.
Line 88: lncRNA acronym needs to be explained in the main text before it is introduced.
Thank you for your kind observation. Acronym has been explained in the immediate text before mentioning it.
Line 176-186: Very long and convoluted sentences. Please rephrase.
Thank you for your comment, we changed and rephrase the wording and separated it into two paragraphs.
Line 349-352: Again, a long and convoluted sentence. Please rephrase.
We appreciate your observation, changes in redaction have been done.
Regarding Table 1: Some references are not relevant to HNSCC because other cancer entities were researched in the References or the relevant lncRNAs are not mentioned. Please recheck the references and modify the relating lncRNA descriptions. Details below:
Thank you for the comment, references regarding other cancer types have been deleted
Reference 39: Study with breast cancer (and not HNSCC).
Thank you for your observation, reference has been deleted
Reference 40: Study without any HNSCC cell lines.
This reference was removed
Reference 41: No mentioning of ANRIL and regarding
We apologize for this mistake, it has been corrected
Reference 51: This review does not mention HNSCC.
Thank you very much for the observation however the authors worked also with oral squamous cell carcinoma (OSCC) which is also part of HNSCC. Nevertheless, an explanation regarding the types of samples has been added to table 1.
Reference 53: Study with cervical cancer.
This reference was removed
Reference 56: Study without HNSCC samples.
This reference was removed
Reference 66: Study with breast epithelial cells and no link with LINC01355.
This reference was removed.
Reviewer 3 Report
Reviewer’s Report on the review entitled “The two faces of immune-related lncRNAs in Head and Neck 2 Squamous Cell Carcinoma” by Urquiza et al., in Cells
The manuscript requires extensive correction both in language and content. The detailed comments are:
1. Extensive language editing is required throughout the manuscript.
2. Lines 87-89: ncRNAs are not miRNAs and lncRNAs. If you intend to mention classification, mention all the subtypes of ncRNAs
3. Line 99: Clarify on what is meant by ‘or their function itself’.
4. Throughout the manuscript sentences are very complex and confusing. Thorough editing is required.
5. Line 242: Check ‘Tfh’ cells. Is it Th cells?
6. Thoroughly check the use of abbreviations. Make sure expansions are abbreviated when they appear for the first time in the manuscript. Also, ensure that all the abbreviations are expanded at least once in the manuscript.
7. Line 268: What is NETosis?
8. The manuscript requires thorough modification in explaining the mechanisms by which different downstream mediators of immune lncRNAs bring about the immune related functions. Most of the instances, the authors have just mentioned the downstream effectors and pointed out that they are involved in immune regulation. Since the review focusses on immune lncRNAs in HNSCC, the authors should explain in detail the exact mechanism of these regulations.
9. It would be appreciable if the authors add a section on drugs or targeted therapies based on lncRNAs or any ncRNAs that is available for numerous HNSCCs in any stages of clinical trials, as a separate section. This would enhance the importance of the immune lncRNAs and how they can contribute to diagnosis, prognosis and therapy of HNSCCs.
Author Response
We acknowledge editors and reviewers for their kind observations regarding our manuscript. All your recommendations have been followed and answered. Changes in the manuscript can be reviewed in the track changes of Word.
Extensive language editing is required throughout the manuscript.
We also appreciate your observations about language editing, the paper has been extensively reviewed and corrected as recommended by Desai et al,. Fluency and grammar have been reviewed and corrected. Missing prepositions and articles were also added improving the text in general. All changes can be seen along the text in Word track changes: “Desai G. How to leverage the language: A guide for medical writers in India. Perspect Clin Res. 2019 Jan-Mar;10(1):37-41. doi: 10.4103/picr.PICR_129_18. PMID: 30834207; PMCID: PMC6371706”. Besides, a thorough textual revision was done by a native English speaker to improve the document
Lines 87-89: ncRNAs are not miRNAs and lncRNAs. If you intend to mention classification, mention all the subtypes of ncRNAs.
Thanks for your kind observation. It has been clarified that according to length ncRNAs can be classified as LncRNA and miRNAs
Line 99: Clarify on what is meant by ‘or their function itself’.
Well noted, thanks for your comment. Sentence has been rephrased.
Throughout the manuscript sentences are very complex and confusing. Thorough editing is required.
We also appreciate your observations about language editing, the paper has been extensively reviewed and corrected as recommended by Desai et al,. Fluency and grammar have been reviewed and corrected. Missing prepositions and articles were also added improving the text in general. All changes can be seen along the text in Word track changes: “Desai G. How to leverage the language: A guide for medical writers in India. Perspect Clin Res. 2019 Jan-Mar;10(1):37-41. doi: 10.4103/picr.PICR_129_18. PMID: 30834207; PMCID: PMC6371706”. Besides, a thorough textual revision was done by a native English speaker to improve the document
Line 242: Check ‘Tfh’ cells. Is it Th cells?
We thank you for the opportune observation, we added the expanded form of the abbreviation.
Thoroughly check the use of abbreviations. Make sure expansions are abbreviated when they appear for the first time in the manuscript. Also, ensure that all the abbreviations are expanded at least once in the manuscript.
We appreciate your comment, the manuscript was reviewed and we made sure to write the expanded form of the abbreviations the first time they appear in the text.
Line 268: What is NETosis?
Thanks for the observation, the clarification of what this process means was made.
The manuscript requires thorough modification in explaining the mechanisms by which different downstream mediators of immune lncRNAs bring about the immune related functions. Most of the instances, the authors have just mentioned the downstream effectors and pointed out that they are involved in immune regulation. Since the review focusses on immune lncRNAs in HNSCC, the authors should explain in detail the exact mechanism of these regulations.
Thank you for your valuable observation, although we would like to be able to explain the exact mechanism by which each lncRNA regulates the conditions in the TME of the HNSCC, to date there are few published studies in this regard. We include the studies available in the databases that provide this information, for example, Jiang, H., et al., M1 macrophage-derived exosomes and their key molecule lncRNA HOTTIP suppress head and neck squamous cell carcinoma progression by upregulating the TLR5/NF-κB pathway. Cell death & disease, 2022. 13(2), 183. doi.org/10.1038/s41419-022-04640-z.
However, most of the papers lack information on the specific mechanism, there are even authors who mention "Future studies should attempt to clarify the mechanisms of how lncRNAs regulate TME in HNSCC": Sun, Q., et al., Identification and Validation of 17-lncRNA Related to Regulatory T Cell Heterogeneity as a Prognostic Signature for Head and Neck Squamous Cell Carcinoma. Frontiers in immunology, 2021. 12, 782216. doi.org/10.3389/fimmu.2021.782216. Both references have been included in the manuscript.
It would be appreciable if the authors add a section on drugs or targeted therapies based on lncRNAs or any ncRNAs that is available for numerous HNSCCs in any stages of clinical trials, as a separate section. This would enhance the importance of the immune lncRNAs and how they can contribute to diagnosis, prognosis and therapy of HNSCCs.
We appreciate your comments, to our knowledge, there are no reports of any therapeutic strategy based on lncRNAs in HNSCC that is in the experimental phase or clinical trial, but considering your recommendation, we add a section on " LncRNAs: therapeutic targets, and clinical relevance in HNSCC" for emphasis on the importance of research in this field.
Round 2
Reviewer 3 Report
The authors have adequately revised the manuscript. I suggest that the revised manuscript may be accepted in the current form.